# Optimal Robustness-Consistency Trade-offs for Learning-Augmented Online Algorithms

**Alexander Wei**
UC Berkeley
awei@berkeley.edu

**Fred Zhang**
UC Berkeley
z0@berkeley.edu

## Abstract

We study the problem of improving the performance of online algorithms by incorporating machine-learned predictions. The goal is to design algorithms that are both *consistent* and *robust*, meaning that the algorithm performs well when predictions are accurate *and* maintains worst-case guarantees. Such algorithms have been studied in a recent line of work initiated by Lykouris and Vassilvitskii (ICML '18) and Kumar, Purohit and Svitkina (NeurIPS '18). They provide robustness-consistency trade-offs for a variety of online problems. However, they leave open the question of whether these trade-offs are tight, i.e., to what extent to such trade-offs are necessary. In this paper, we provide the first set of non-trivial lower bounds for competitive analysis using machine-learned predictions. We focus on the classic problems of ski rental and non-clairvoyant scheduling and provide optimal trade-offs in various settings.

# 1 Introduction

The vast gains in predictive ability by machine learning models in recent years have made them an attractive approach for algorithmic problems under uncertainty: One can train a model to predict outcomes on historical data and then respond according to the model's predictions in future scenarios. For example, when renting cloud servers, a company might need to decide whether to pay on-demand or reserve cloud servers for an entire year. The company could try to optimize their purchasing based on a model learned from past demand. However, a central concern for applications like these is the lack of provable bounds on worst-case performance. Modern machine learning models may produce predictions that are embarrassingly inaccurate (e.g., [SZS+14]), especially when trying to generalize to unfamiliar inputs. The potential for such non-robust behavior is be problematic in practice, when users of machine learning-based systems desire at least some baseline level of performance in the worst case.

On the other hand, the algorithms literature has long studied algorithms with worst-case guarantees. In particular, the theory of *online algorithms* focuses on algorithms that perform well under uncertainty, even when inputs are chosen adversarially. A key metric in this literature is the *competitive ratio*, which is the ratio between the worst-case performance of an algorithm (without knowledge of the future) and that of an offline optimal algorithm (that has full knowledge of the future).[1] That is, an algorithm with a competitive ratio of $\mathcal{C}$ does at most $\mathcal{C}$ times worse than any other algorithm, even in hindsight. The classical study of online algorithms, however, focuses on the worst-case outcome over all possible inputs. This approach can be far too pessimistic for many real-world settings, leaving room for improvement in more optimistic scenarios where the algorithm designer has some prior signal about future inputs.

Recent works by Lykouris and Vassilvitskii [LV18] and Kumar, Purohit and Svitkina [KPS18] intertwine these two approaches to algorithms under uncertainty by augmenting algorithms with machine learned predictions. The former studies the online caching problem, whereas the latter focuses on the classical problems of ski-rental and non-clairvoyant job scheduling. They design algorithms that (1) perform excellently when the prediction is accurate and (2) have worst-case guarantees in the form of competitive ratios. For such augmented online algorithms, they introduce the metrics of *consistency*, which measures the competitive ratio in the case where the machine learning prediction is perfectly accurate, and *robustness*, which is the worst-case competitive ratio over all possible inputs. Moreover, the algorithms they design have a natural *consistency-robustness trade-off*, where one can improve consistency at the cost of robustness and vice versa. These two works, however, do not discuss the extent to which such trade-offs are necessary, *i.e.*, whether the given trade-offs are tight.[2]

In this work, we provide the first set of optimal results for online algorithms using machine-learned predictions. Our results are the following:

(i) For the ski-rental problem, we give tight lower bounds on the robustness-consistency trade-off in both deterministic and randomized settings, matching the guarantees of the algorithms given by [KPS18].

(ii) For the non-clairvoyant job scheduling problem, we provide a non-trivial lower bound that is tight at the endpoints of the trade-off curve.

   Moreover, for the case of two jobs, we give matching upper and lower bounds on the full trade-off. The algorithm improves significantly upon that of [KPS18].

Conceptually, our results show that merely demanding good performance under perfect prediction can require substantial sacrifices in overall robustness. That is, this trade-off between good performance in the ideal setting and overall robustness is deeply intrinsic to the design of learning-augmented online algorithms.

## 1.1 Our results

**Ski-rental.** The *ski rental problem* is a classical online algorithms problem [KMRS88] with a particularly simple model of decision-making under uncertainty. In the problem, there is a skier who is out to ski for an *unknown* number of days. The first morning, the skier must either rent skis for a cost of $1 or buy skis for a cost of $B. Each day thereafter, the skier must make the same decision again as long as she has not yet purchased skis. The goal for the skier is to follow a procedure that minimizes competitive ratio. Variations of the ski-rental problem have been used to model a diverse set of scenarios, including snoopy caching [KMRS88], dynamic TCP acknowledgement [KKR03], and renting cloud servers [KKP13].

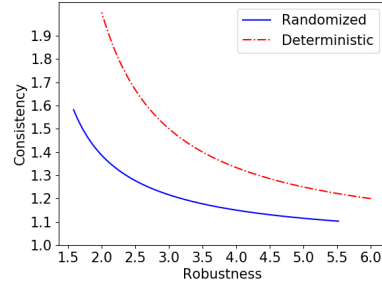

Figure 1: Tight deterministic and randomized trade-offs for learning-augmented ski-rental.

In our setting of ski-rental with a machine-learned prediction, we assume that, in addition to knowing $B$, the skier has access to a prediction $y$ for the number of days she will ski. Let $\eta$ denote the absolute error of the prediction $y$ (i.e., if she actually skis for $x$ days, then $\eta = |x - y|$). Furthermore, define $c(\eta)$ to be the skier's worst-case competitive ratio over all $y$ given $\eta$. We say that the procedure is $\gamma$-*robust* if $c(\eta) \leq \gamma$ for any $\eta$ and that it is $\beta$-*consistent* if $c(0) \leq \beta$. We prove deterministic and randomized lower bounds on the robustness-consistency trade-off that match the algorithmic results in [KPS18]. Specifically, we show:

**Theorem 1.1** (Deterministic Lower Bound for Ski-Rental; also in [GP19, ADJ$^+$20])**.** Let $\lambda \in (0, 1)$ be a fixed parameter. Any $(1 + \lambda)$-consistent deterministic algorithm for ski-rental with machine-learned prediction problem is at least $(1 + 1/\lambda)$-robust.

We remark that this deterministic bound is simple to prove and has also appeared in two prior works [GP19, ADJ$^+$20].

**Theorem 1.2** (Randomized Lower Bound for Ski-Rental)**.** Any (randomized) algorithm for ski-rental with machine-learned prediction that achieves robustness $\gamma$ must have consistency

$$\beta \geq \gamma \log \left( 1 + \frac{1}{\gamma - 1} \right).$$

In particular, any (randomized) algorithm achieving robustness $\gamma \leq 1/(1 - e^{-\lambda})$ for the ski-rental with machine-learned prediction problem must have consistency $\beta \geq \lambda/(1 - e^{-\lambda})$.

**Non-clairvoyant scheduling.** The *non-clairvoyant job scheduling problem* was first studied in an online setting by Motwani, Phillips, and Torng [MPT94]. This problem models scheduling jobs on a single processor, where the jobs have unknown processing times and the objective is to minimize the completion time (*i.e.*, the sum of the job completion times). More formally, the algorithm initially receives $n$ job requests with *unknown* processing times $x_1, x_2, \cdots, x_n$ and is asked to schedule them on a single machine, allowing for preemptions. If the completion time of job $i$ is $t_i$, then the total *completion time* of the algorithm is $\sum_{i=1}^{n} t_i$.

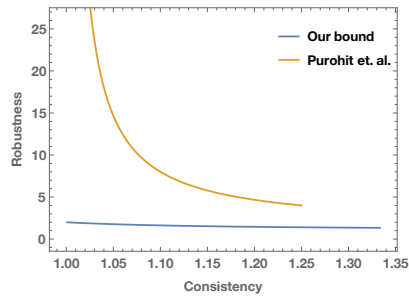

Figure 2: Tight trade-offs for scheduling two jobs

In the learning-augmented version of the problem, we additionally provide the algorithm with predictions $y_1, y_2, \cdots, y_n$ of the processing times $x_1, x_2, \cdots, x_n$. Let $\eta = \sum_i |x_i - y_i|$ be the $\ell_1$ error of the prediction and $c(\eta)$ be the algorithm's worst-case competitive ratio given $\eta$. As before, we say an algorithm is $\gamma$-robust if $c(\eta) \leq \gamma$ for any $\eta$ and $\beta$-consistent if $c(0) \leq \beta$. Our first result is a lower bound on the robustness-consistency trade-off in the general case.

**Theorem 1.3** (Lower bound for non-clairvoyant scheduling with $n$ jobs). Any $(1 + \lambda)$-consistent algorithm for non-clairvoyant scheduling with machine-learned prediction must have robustness

$$\gamma \geq \frac{n + n(n+1)\lambda}{1 + \lambda(n+1)(n+2)/2}.$$

This bound is tight at the endpoints of the trade-off. When $\lambda = 0$, we have $c(\eta) \geq n$, which is achieved by any (non-idling) algorithm. On the other hand, when $\lambda = 1 - \frac{2}{n+1}$ (so $1 + \lambda = 2 - \frac{2}{n+1}$), we have $c(\eta) \geq 2 - \frac{2}{n+1}$, which is the tight bound of [MPT94] (achieved by round-robin).[3]

On the other hand, Kumar, Purohit and Svitkina [KPS18] give an algorithm that is $(1 + \lambda)/2\lambda$-consistent and $2/(1 - \lambda)$-robust for $\lambda \in (0, 1)$. In the case of $n = 2$, the robustness can be improved to $4/(3 - 3\lambda)$. We provide a significantly better trade-off (Figure 2) and a matching lower bound in this regime. Our algorithm is 2-competitive over all parameter choices, while their algorithm has robustness tends to infinity as consistency goes to 1.

**Theorem 1.4** (Tight bound for non-clairvoyant scheduling of 2 jobs). In the case of 2 jobs, there is an algorithm that achieves $(1 + \lambda)$-consistency and $(1 + 1/(1 + 6\lambda))$-robustness for non-clairvoyant scheduling with machine-learned prediction, for any $\lambda \in (0, 1/3)$.[4] Moreover, this bound is tight.

## 1.2 Related work

For learning-based ski-rental, the result of [KPS18] has since been extended by [LHL19, GP19]. Scheduling with predictions is also studied by [LLMV20, Mit20, Mit19], though under different prediction models or problem settings. The results of [LV18] on online caching with ML predictions have been improved and generalized by [ACE+20, Roh20, JPS20, Wei20]. Several other learning-augmented online problems have also been considered in the literature, including matching, optimal auctions and bin packing [DH09, KPS+19, MV17, AGKK20, ADJ+20].

Online algorithms (without ML predictions) are a classical subject in the algorithms literature. The (classic) ski-rental problem is well-understood: It is known that there exists a 2-competitive deterministic algorithm [KMRS88]. This can be further improved to $e/(e - 1)$ using randomization and is known to be optimal [KMMO94]. There are also numerous extensions of the problem, including snoopy caching [KMRS88] and dynamic TCP acknowledgment [KKR03]. The non-clairvoyant scheduling problem was first studied by [MPT94]. They show that for $n$ jobs the round-robin heuristic achieves a competitive ratio of $2 - 2/(n + 1)$ and provide a matching lower bound. They also show that randomization provides at most a minor lower-order improvement to the competitive ratio. Our work revisits these classical results by extending their lower bounds to settings where we want to optimize for consistency (with respect to a prediction) in addition to worst-case competitive ratio.

Another related line of inquiry is the study of online problems in stochastic settings, where the inputs come from certain distribution [HB09, FMMM09, MNS12, MGZ12, Mit20, EKM18]. We note that this model differs from ours in that we do not make any assumptions on the distribution or stochasticity of inputs.

Finally, using machine learning to design algorithms under uncertainty has been explored in other settings as well, such as online learning [KLMS19, BCKP20] and data streams [HIKV19, AIV19, JLL+20, CGP20]. A number of works also study learning-based methods for numerical linear algebra, combinatorial optimization, and integer programming [BPL+16, KDZ+17, BDSV18, NOST18, LCK18, KvHW19, SLB+19, AMW19, CGT+20, IVY19, AKL+19, DGE+19].

## 1.3 Preliminaries and notations

In our analyses that follow, we use ALG to denote the cost incurred by the algorithm on a given input and prediction. We use OPT to denote the optimal cost achievable by an algorithm with full

knowledge of the future (i.e., an offline algorithm). Note that ALG is a function of the input and the prediction, while OPT depends only on the input. The competitive ratio for a given input and prediction is simply the ratio ALG/OPT.

In terms of this notation, an algorithm is $\beta$-consistent if ALG/OPT $\leq \beta$ for all situations where the input is the same as the prediction; an algorithm is $\gamma$-robust if ALG/OPT $\leq \gamma$ for all pairs of input and prediction.

## 2 Ski Rental

In the ski-rental problem, a skier is out on a ski trip, which will end after the $x$-th day for some *unknown* $x$. Each day, if she has not yet bought skis, the skier may either rent skis for \$1 or buy skis for \$$B$ and ski for free from then on. In the learning-augmented version of the problem, the skier is also provided with a machine-learned prediction $y$ of $x$ that she may use to aid her decision.

We first state the algorithmic results of Kumar, Purohit and Svitkina [KPS18], which we will prove to be optimal. Their algorithms require a hyperparameter $\lambda \in (0,1)$ that dictates the trade-off between robustness and consistency. Given $\lambda$, the deterministic and randomized algorithms of [KPS18] for ski rental with machine-learned predictions proceed as follows:

```
Deterministic-Ski-Rental(y, B):
    If y ≥ B,
        Buy at the start of day ⌈λB⌉.
    Otherwise,
        Buy at the start of day ⌈B/λ⌉.
```

```
Randomized-Ski-Rental(y, B):
    If y ≥ B,
        Let k = ⌈λB⌉.
    Otherwise,
        Let k = ⌈B/λ⌉.
    Select day i ∈ [k] with probability proportional to (1 − 1/B)^(k−i).
    Buy at the start of day i.
```

Kumar, Purohit and Svitkina [KPS18] show that the algorithms achieve the following robustness-consistency trade-offs:

**Theorem 2.1** (Theorem 2.2 of [KPS18]). *Given a parameter $\lambda \in (0,1)$, the* `Deterministic-Ski-Rental` *algorithm is $(1 + 1/\lambda)$-robust and $(1 + \lambda)$-consistent.*

**Theorem 2.2** (Theorem 2.3 of [KPS18]). *Given a parameter $\lambda \in (0,1)$, the* `Randomized-Ski-Rental` *algorithm is $\left(\frac{1}{1-e^{-(\lambda-1/B)}}\right)$-robust and $\left(\frac{\lambda}{1-e^{-\lambda}}\right)$-consistent.*

Notice that for large $B$, our randomized lower bound (Theorem 1.2) essentially matches the guarantee of Theorem 2.2.

### 2.1 Deterministic Lower Bound

In this section, we prove Theorem 1.1, which also appeared in [GP19, ADJ$^+$20]. Since the algorithm is deterministic, we proceed by a an adversarial argument. Let $x$ be the last day of the ski season. The high-level idea is to fix a specific $y$, and then consider two instances, one where $x = y$ and one where $x \neq y$. Since the algorithm does not know $x$, it cannot distinguish between these two cases and therefore must output a unique day $t$ (for purchasing skis) given $y$ and $B$. Suppose $y$ is large, say, greater than $B$. Then, intuitively, $t$ must be fairly small to satisfy consistency. Given this constraint, in the other instance, we let the adversary choose an $x \neq y$ that yields the worst possible competitive ratio. We will show that this competitive ratio indeed matches the robustness upper bound.

*Proof of Theorem 1.1.* Let $y$ be the prediction and $\eta = |y - x|$ be the error. Consider a deterministic algorithm that achieves $(1 + \lambda)$ consistency. Suppose $y > (1 + \lambda)B$, and let $t$ be the day on which the algorithm purchases skis (given $y$ and $B$).

First, suppose $t \geq y$. When $x = y$, we have OPT $= B$ and ALG $= y$. Then the competitive ratio is $y/B$, which must be bounded by $1 + \lambda$ by our consistency requirement, but this contradicts the

assumption $y > (1 + \lambda)B$. Second, suppose $B < t < y$. Again, when $x = y$, $\mathsf{OPT} = B$, and $\mathsf{ALG} = t + B - 1$. By the $(1 + \lambda)$-consistency, $(t + B - 1)/B \leq 1 + \lambda$. Thus, $(t - 1)/B \leq \lambda < 1$, contradicting the assumption that $t > B$. Therefore, simply to achieve $(1 + \lambda)$-consistency, the algorithm must output $t \leq B$. Now under this condition, we consider two cases. We use the case when $y = x$ to derive a bound on $\lambda$, and apply this along with an adversarial argument in the case when $y \neq x$ to obtain our robustness lower bound.

(i) Suppose $x = y$. Since $y > B$, we have $\mathsf{OPT} = B$. On the other hand, $\mathsf{ALG} = t + B - 1$, as $t < x$. Thus, the algorithm does $1 + (t - 1)/B$ times worse than optimal. Assuming that the algorithm is $(1 + \lambda)$-consistent, we have $1 + (t - 1)/B \leq 1 + \lambda$, so $t \leq \lambda B + 1$.

(ii) Suppose $x \neq y$. We adversarially set $x = t$; note that $x \leq B$. Thus, $\mathsf{OPT} = x = t$ and $\mathsf{ALG} = t + B - 1$. Our bound on $t$ from (i) now lower bounds the competitive ratio as $(t + B - 1)/t \geq 1 + (B - 1)/(\lambda B + 1)$. For large $B$, this lower bound approaches $1 + 1/\lambda$. This shows that $c(\eta) \geq 1 + 1/\lambda$ and thus completes the proof. $\qquad\square$

## 2.2 Randomized Lower Bound

The starting point of our randomized lower bound is the well-known fact that the ski-rental problem can be expressed as a linear program (see, *e.g.*, [BN09]). Our key observation then is that the consistency and robustness constraints are in fact also linear. Somewhat surprisingly, we show that the resulting linear program can be solved *analytically* in certain regimes. By exploiting the structure of the linear program, we will determine the optimal robustness for any fixed consistency, and this matches the trade-off given by Theorem 2.2 (when $y \gg B$ and for large $B$).

The proof of our randomized lower bound (Theorem 1.2) is fairly technical. Thus, we defer the proof to Appendix A and only present a sketch here.

*Proof sketch of Theorem 1.2.* As a first step, we can characterize algorithms for ski rental as feasible solutions to an infinite linear program, with variables $\{p_i\}_{i \in \mathbb{N}}$ indicating the probability of buying at day $i$. The constraints of robustness and consistency can be written as linear constraints on this representation. Given $\gamma$ and $\beta$, understanding whether a $\gamma$-robust and $\beta$-consistent algorithm exists therefore reduces to checking if this linear program is feasible. (In particular, we do not have an objective for the linear program.)

First, we ask that the $p_i$'s define a probability distribution. That is, $p_i \geq 0$ and

$$\sum_{i=1}^{\infty} p_i = 1. \tag{2.1}$$

Second, to satisfy the consistency constraint, the algorithm must have expected cost within $\beta \cdot \mathsf{OPT}$ when $y = x$. In this case, the ski season ends at $i = y$, so there is no additional cost afterwards.

$$\sum_{i=1}^{y}(B + i - 1)p_i + y \sum_{i=y+1}^{\infty} p_i \leq \beta \min\{B, y\}. \tag{2.2}$$

Third, each value of $x$ gives a distinct constraint for robustness, where the left side is the expected cost and the right side is $\gamma \cdot \mathsf{OPT}$. When $x \leq B$, $\mathsf{OPT} = x$, so we have

$$\sum_{i=1}^{x}(B + i - 1)p_i + x \sum_{i=x+1}^{\infty} p_i \leq \gamma x \quad \forall x \leq B. \tag{2.3}$$

If $x > B$, then $\mathsf{OPT} = B$. The robustness constraints are infinitely many, given by

$$\sum_{i=1}^{x}(B + i - 1)p_i + x \sum_{i=x+1}^{\infty} p_i \leq \gamma B \quad \forall x > B. \tag{2.4}$$

Having set up this LP, the remainder of the proof follows in two steps. First, we show that this (infinite) LP can be reduced to a finite one with $B + 1$ constraints and $y$ variables. We then proceed to analytically understand the solution to the LP. This allows us to lower bound the parameter $\gamma$ given any $\beta$, and it indeed matches the upper bound given by [KPS18]. $\qquad\square$

# 3 Non-clairvoyant Scheduling

In the non-clairvoyant scheduling problem, we have to complete $n$ jobs of unknown lengths $x_1, x_2, \cdots, x_n$ using a single processor. The processor only learns the length of a job upon finishing that job. The goal in this problem is to schedule the jobs with preemptions to minimize the total completion time, *i.e.*, the sum of the times at which each job finishes. Observe that no algorithm can achieve a non-trivial guarantee if preemptions are disallowed. The problem has been well-studied in the classic setting. Motwani, Phillips, and Torng [MPT94] show that the *round-robin* (RR) algorithm achieves $2 - 2/(n+1)$ competitive ratio, which is the best possible among deterministic algorithms. The algorithm simply assigns a processing rate of $1/k$ to each of the $k$ unfinished jobs at any time. (Note that since preemption is allowed, we can ease our exposition by allowing concurrent jobs run on the processor, with rates summing to at most 1.)

Now, suppose one has access to a machine-learned oracle that produces predictions $y_1, y_2, \cdots, y_n$ of the processing times $x_1, x_2, \cdots, x_n$. Define $\eta = \sum_i |x_i - y_i|$ to be the total prediction error. We would like to design algorithms that achieve a better competitive ratio than $2 - 2/(n+1)$ when $\eta = 0$ and while preserving some constant worse-case guarantee.

## 3.1 A General Lower Bound

Our first result is a lower bound on the robustness-consistency trade-off that is tight at the endpoints of the trade-off curve. Note that since the classic work [MPT94] provides a $c = 2 - 2/(n+1)$ competitive ratio (with no ML prediction), one can always achieve $c$-robustness and $c$-consistency simultaneously. Hence, as we remarked, Theorem 1.3 is tight at $\lambda = 0$ and $\lambda = 1 - \frac{2}{n+1}$. We now prove the theorem.

*Proof of Theorem 1.3.* Consider an algorithm that achieves $1 + \lambda$ consistency. Let the predictions be $y_1 = y_2 = \cdots = y_n = 1$. Let $d(i, j)$ denote the amount of processing time on job $i$ before job $j$ finishes. Assume without loss of generality that job 1 is the first job to finish and that when it finishes, we have $d(i, i) \geq d(j, j)$ for all $i < j$. Consistency requires

$$(1 + \lambda) \cdot \mathsf{OPT} = \frac{n(n+1)}{2}(1 + \lambda) \geq \sum_{i,j} d(i,j) + \sum_{i=2}^{n}(n - i + 1)(1 - d(i,i)),$$

where the first term represents the costs incurred thus far, and the second term represents the minimum cost required to finish from this state. Simplifying, we obtain the condition

$$\frac{n(n+1)}{2}\lambda \geq \sum_{i=2}^{n}(i - 1) \cdot d(i,i), \tag{3.1}$$

as $d(i, j) = d(i, i)$ for all $i$ at this point in the execution.

Now, consider a (adversarial) setup with $x_i = d(i, i) + \varepsilon$, where we take $d(i, i)$ to be as measured upon the completion of job 1 and $\varepsilon > 0$ to be a small positive number. For this instance, we have

$$\mathsf{OPT} = 1 + \sum_{i=2}^{n} i x_i + O(\varepsilon).$$

We also have, based on the execution of the algorithm up to the completion of job 1, that

$$\mathsf{ALG} \geq n\left(1 + \sum_{i=2}^{n} x_i\right).$$

To show a consistency-robustness lower bound, it suffices to lower bound $\mathsf{ALG}/\mathsf{OPT}$ subject to the consistency constraint. Equivalently, we can upper bound

$$\frac{\mathsf{OPT}}{\mathsf{ALG}} - \frac{1}{n} \leq \frac{1}{n}\left(\frac{1 + \sum_{i=2}^{n}(i-1)x_i + O(\varepsilon)}{1 + \sum_{i=2}^{n} x_i}\right).$$

Suppose we know a priori that the value of the numerator is $C + 1 + O(\varepsilon)$ (i.e., $\sum_{i=2}^{n}(i-1)x_i = C$). To maximize the quantity on the right-hand side, we would want to have $\sum_{i=2}^{n} x_i$ be as small as possible subject to the constraints that $x_i \geq x_j \geq 0$ if $i < j$ and

$$\sum_{i=2}^{n}(i-1)x_i = C.$$

Observe that this optimization problem is a linear program. For this linear program, suppose we have a feasible solution with $x_i > x_{i+1}$. Such a solution cannot be optimal, as we can set $x_i \leftarrow x_i - \frac{\alpha}{i-1}$ and $x_{i+1} \leftarrow x_{i+1} + \frac{\alpha}{i}$ for sufficiently small $\alpha > 0$, reducing the objective while remaining feasible. Thus, if an optimal solution exists, it must have $x_2 = x_3 = \cdots = x_n$. It is not hard to see that this linear program is bounded and feasible, so an optimum does exist. It follows that for a given $C$, we want to set $x_2 = x_3 = \cdots = x_n = \frac{2C}{n(n-1)}$, in which case the right-hand side is equal to

$$\frac{C + 1 + O(\varepsilon)}{1 + \frac{2C}{n}} - \frac{n-1}{2} + \frac{n-1}{2} = \frac{-\frac{n-3}{2} + O(\varepsilon)}{1 + \frac{2C}{n}} + \frac{n-1}{2}.$$

To maximize the leftmost term, which has a negative numerator (for sufficiently small $\varepsilon$), we want to maximize $C$. However, we know from (3.1) that $C = \sum_{i=2}^{n}(i-1)x_i \leq \frac{n(n+1)}{2}\lambda$. Therefore, we have the upper bound

$$\frac{\mathsf{OPT}}{\mathsf{ALG}} - \frac{1}{n} \leq \frac{1}{n}\left(\frac{\frac{n(n+1)}{2}\lambda + 1 + O(\varepsilon)}{1 + (n+1)\lambda}\right).$$

Finally, taking $\varepsilon \to 0$ yields the desired bound

$$\frac{\mathsf{ALG}}{\mathsf{OPT}} \geq \frac{n + n(n+1)\lambda}{1 + \frac{(n+1)(n+2)}{2}\lambda}. \qquad \square$$

## 3.2 A Tight Complete Trade-off for Two Jobs

We now consider the special case of having $n = 2$ jobs. It is always possible to achieve $4/3$ competitiveness by round-robin [MPT94], and with machine-learned predictions, Kumar, Purohit, and Svitkina [KPS18] proves an $(1 + \lambda)/2\lambda$-consistency and $4/(3 - 3\lambda)$-robustness trade-off. We show that this trade-off can be significantly improved and that our new bound is in fact tight.

**Lower bound.** We start by proving our lower bound. Here, we remark that any lower bound for $k$ jobs directly implies the same lower bound for any $n \geq k$ jobs, since one can add $n - k$ dummy jobs with 0 predicted and actual processing times. Thus, the lemma below also holds for $n > 2$.

**Lemma 3.1** (Lower bound for non-clairvoyant scheduling). *For the non-clairvoyant scheduling problem of 2 jobs, any algorithm that achieves $(1 + \lambda)$-consistency must be at least $1 + (1/(1 + 6\lambda))$-robust for a $\lambda \in (0, 1/3)$.*

*Proof.* Consider a $(1 + \lambda)$-consistent algorithm $\mathcal{A}$. Suppose the inputs are predictions $y_1 = y_2 = 1$. First, we focus on an instance $I$, where $x_1 = y_1, x_2 = y_2$. Let $d(i, j)$ denote the amount of processing time on job $i$ before job $j$ finishes for this instance, and assume without loss of generality that the algorithm finishes job 1 first. Observe in this scenario the consistency requirement asks that $\mathcal{A}$ must produce a schedule with total completion time at most $(1 + \lambda)(2y_1 + y_2) = 3 + 3\lambda$. As job 1 finishes first, $d(1, 2) = 1$. Since $x_1 = x_2 = 1$ and $\mathsf{ALG} = x_1 + x_2 + d(1, 2) + d(2, 1)$, we must have

$$d(2, 1) \leq \lambda(2y_1 + y_2) = 3\lambda. \tag{3.2}$$

Now we consider an adversarial instance $I'$ with same predictions ($y_1 = y_2 = 1$), but different choices of actual processing times. In particular, let $x_1 = 1$ but $x_2 = d(2, 1) + \epsilon$ for an infinitesimal constant $\epsilon$. Since the inputs to the algorithm are the same as in the previous instance $I$, it would start off by producing the same schedule. In particular, the algorithm would finish job 1 first at time $1 + d(2, 1)$, then finish job 2 immediately afterwards. Therefore,

$$\mathsf{ALG} = 2 + 2d(2, 1) + \epsilon. \tag{3.3}$$

On the other hand, since $\lambda \leq 1/3$, $x_2 \leq x_1$, we have

$$\mathsf{OPT} = 2x_2 + x_1 = 2d(2, 1) + 2\epsilon + 1. \tag{3.4}$$

By (3.2), we get that the competitive ratio is at least $1 + 1/(1 + 6\lambda)$ as $\epsilon \to 0$. $\qquad \square$

**Upper bound.** To complete the proof of Theorem 1.4. We show that the algorithm from [KPS18] can be improved. Our new scheduling scheme proceeds in two stages. First, it follows the round-robin algorithm until the consistency constraint is tight. Then, it processes jobs in a greedy order, starting with the job of minimum prediction time. We name the algorithm `Two-Stage-Schedule` and prove the following guarantee:

**Lemma 3.2** (Algorithm for non-clairvoyant scheduling). *For the non-clairvoyant scheduling problem of 2 jobs, the algorithm* `Two-Stage-Schedule` *achieves* $(1 + \lambda)$-*consistency and* $(1 + 1/(1 + 6\lambda))$-*robustness for a* $\lambda \in (0, 1/3)$.

The proof can be found in Appendix B. Finally, combining Lemma 3.2 and Lemma 3.1 proves Theorem 1.4.

## 4    Conclusion

In this paper, we give lower bounds for the learning-augmented versions of the ski-rental problem and non-clairvoyant scheduling. In doing so, we show that robustness-consistency trade-offs are deeply intrinsic to the design of online algorithms that are robust in the worst case yet perform well when machine-learned predictions are accurate.

A broad future direction is to use our techniques to investigate tight robustness-consistency trade-offs for other learning-augmented online algorithms (e.g., online matching or generalizations of ski-rental) following the spate of recent works on this topic.

## Broader Impact

Our work is on the foundations of learning-augmented online algorithms. In particular, we focus on the fundamental limitations and trade-offs inherent to the approach (as opposed to providing new algorithms). Since our work is primarily theoretical, we believe that the immediate impact of our work will be on the academic design of algorithms within this budding research area. More broadly, we hope that research on this topic will lead to safer and more robust applications of ML in decision-making settings.

## Acknowledgments and Disclosure of Funding

We would like to thank Constantinos Daskalakis, Piotr Indyk, and Jelani Nelson for their comments on drafts of this paper.

## Footnotes

[1]For randomized algorithms, one considers the *expected* competitive ratio, which compares the expected cost (taken over the randomness of the algorithm) to the cost of the offline optimal.

[2]We remark that Lykouris and Vassilvitski [LV18] prove for the online caching problem that one can achieve constant competitiveness under perfect predictions while having $O(\log k)$ competitiveness in the worst case. This bound is optimal up to constant factors by the classical algorithms for online caching [FKL+91].

[3] A competitive ratio of $2 - 2/(n + 1)$ can always be achieved (even without ML predictions) [MPT94], so we do not need to consider consistency $1 + \lambda$ for $\lambda \geq 1 - 2/(n + 1)$

[4] Kumar, Purohit and Svitkina [KPS18] uses large $\lambda$ to indicate low consistency, whereas we use small $\lambda$ for low consistency. The results are comparable up to a reparametrization. Also, round-robin has a competitive ratio of $4/3$ for 2 jobs (without using predictions) [MPT94], so we do not need to consider consistency $1 + \lambda$ for $\lambda \geq 1/3$.

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
