[Supplementary Material]

# A  Proof of Theorem 1.2

**LP Construction**  First, consider the following LP construction for the learning-augmented ski-rental problem: We use $\gamma$ to denote the robustness parameter and $\beta$ the consistency parameter. We assume without loss of generality that $\beta < \gamma$; otherwise, the consistency requirement is redundant. Consider a infinite LP, with variables $\{p_i\}$ indicating the probability of buying at day $i$. First, we ask that the $p_i$'s define a probability distribution. That is, $p_i \geq 0$ and

$$\sum_{i=1}^{\infty} p_i = 1 \qquad \text{(probability distribution)}$$

Second, to satisfy the consistency constraint, the algorithm must have expected cost within $\beta \cdot \mathsf{OPT}$ when $y = x$. In this case, the ski season ends at $i = y$, so there is no additional cost afterwards.

$$\sum_{i=1}^{y}(B + i - 1)p_i + y \sum_{i=y+1}^{\infty} p_i \leq \beta \min\{B, y\}. \qquad \text{(consistency)}$$

Third, each value of $x$ gives a distinct constraint for robustness, where the left side is the expected cost and the right side is $\gamma \cdot \mathsf{OPT}$. When $x \leq B$, $\mathsf{OPT} = x$, so we have

$$\sum_{i=1}^{x}(B + i - 1)p_i + x \sum_{i=x+1}^{\infty} p_i \leq \gamma x \quad \forall x \leq B \qquad (\text{A.1})$$

If $x > B$, then $\mathsf{OPT} = B$. The robustness constraints are infinitely many, given by

$$\sum_{i=1}^{x}(B + i - 1)p_i + x \sum_{i=x+1}^{\infty} p_i \leq \gamma B \quad \forall x > B \qquad (\text{A.2})$$

We remark that for each $p_i$, its coefficient is non-decreasing as we go down. We denote the robustness constraint corresponding to $x$ by $\mathcal{C}(x)$ and the entire (infinite) LP by $\mathcal{P}$.

From now on, we focus on the case when $y \geq 2B - 1$. In this case, the consistency constraint is

$$\sum_{i=1}^{y}(B + i - 1)p_i + y \sum_{i=y+1}^{\infty} p_i \leq \beta B. \qquad \text{(consistency')}$$

**Reducing to a Finite LP**  We will show that $\mathcal{P}$ can be reduced to be a finite LP of $B+1$ constraints and $y$ variables.

**Lemma A.1.** If $y \geq B$, the robustness constraints between $\mathcal{C}(B)$ and $\mathcal{C}(y)$ are redundant.

*Proof.*  Observe that these constraints are dominated by the constraint consistency', since $\beta \leq \gamma$ and their left sides are bounded by the left side of consistency'. $\qquad \square$

We start by identifying redundant variables in $\mathcal{P}$.

**Lemma A.2** (Redundant variables). If $y \geq 2B - 1$ and $\mathcal{P}$ is feasible, then there exists a feasible solution to $\mathcal{P}$ such that $p_i = 0$ for all $i \geq y + 1$.

*Proof.*  Let $p$ be a feasible solution to $\mathcal{P}$. First, we argue that there exists a feasible solution $p'$ such that $p'_{y+2} = p'_{y+3} = \cdots = 0$. To eliminate $p_{y+2}$, consider $p'$ where $p'_{y+2} = 0$, $p'_{y+1} = p_{y+1} + p_{y+2}$, and $p'_i = p_i$ for $i \notin \{y+1, y+2\}$. Clearly, $p'$ still defines a probability distribution. We now check $p'$ is feasible.

  (i) Since $y \geq B$, the consistency constraint is satisfied.

  (ii) The robustness constraints from $\mathcal{C}(1)$ to $\mathcal{C}(B-1)$ are satisfied, by the coefficients in these constraints.

(iii) By Lemma A.1, we focus on the robustness constraints from $\mathcal{C}(y+1)$. First, the coefficient of $p_{y+2}$ is greater than the coefficient of $p_{y+1}$ in the constraint $\mathcal{C}(i)$. for all $i \geq y+2$. Hence, $p'$ satisfies these constraints. Then, note that the constraint $\mathcal{C}(y+1)$ is dominated by the constraint $\mathcal{C}(y+2)$, and thus $p'$ satisfies it.

Applying this argument iteratively, we can eliminate all variables $p_i$ for $i \geq y+2$. Now it is easy to observe that constraints $\mathcal{C}(j)$ for $j \geq y+2$ are redundant.

Finally, to eliminate $p_{y+1}$, consider $p'_{y+1} = 0, p'_{y-B+1} = p_{y-B+1} + p_{y+1}$, and $p'_i = p_i$ for $i \notin \{y-B+1, y+1\}$. Observe that since $y \geq 2B-1$,

(i) in the consistency' constraint and constraint $\mathcal{C}(y+1)$, the coefficient of $p_{y+1}$ and $p_{y-B+1}$ are both $y$;

(ii) in the constraints between $\mathcal{C}(1)$ and $\mathcal{C}(B-1)$, the coefficient of $p_{y+1}$ and $p_{y-B+1}$ are both $B$.

It follows that all constraints are satisfied by $p'$. $\qquad\square$

**Corollary A.3** (Redundant constraints). If $y \geq 2B-1$, all robustness constraints in $\mathcal{P}$ are redundant except those between $\mathcal{C}(1)$ and $\mathcal{C}(B-1)$.

*Proof.* This follows directly from the definition of the robustness constraints and Lemma A.2. $\quad\square$

**Lemma A.4** (Tight constraints). If $\mathcal{P}$ is feasible, there exists a solution such that for all $1 < k < y$, if $p_k > 0$, then the preceding robustness constraints $\mathcal{C}(k')$ for $1 \leq k' < k$ are all tight.

*Proof.* Using the probability distribution constraint, we can rewrite each robustness constraint $\mathcal{C}(i)$ as

$$(B-i)p_1 + (B-i+1)p_2 + \cdots + (B-1)p_i \leq \gamma \min\{B, i\} - i. \tag{A.3}$$

Let $p$ be a feasible solution. First, we claim that when shifting probability mass from $p_k$ to $p_{k-1}$, the slack for all robustness constraints is non-decreasing, except for $\mathcal{C}(k-1)$. Note that

(i) for $k' \geq k$, since the coefficient of $p_{k-1}$ in $\mathcal{C}(k')$ is less than that of $p_k$, we strictly increase the slack; and

(ii) for $k' < k-1$, $\mathcal{C}(k')$ has no dependence on either $p_k$ or $p_{k-1}$, so the slack remains unchanged.

Second, we claim that if $\mathcal{C}(k-1)$ has non-zero slack, then $p_k > 0$ or $\mathcal{C}(k)$ has non-zero slack. Indeed, if $p_k = 0$, then constraint $\mathcal{C}(k-1)$ is stronger than constraint $\mathcal{C}(k)$. Since constraint $\mathcal{C}(k-1)$ has non-zero slack, constraint $\mathcal{C}(k)$ must also have non-zero slack.

Let $s = (s_1, s_2, \ldots, s_{y-1})$ be the vector of slacks for the robustness constraints $\mathcal{C}(i)$. The above two claims together show that if a feasible solution is such that $p_k > 0$, but one of the preceding robustness constraints is not tight, then we can shift some probability mass so that the robustness slack vector becomes lexicographically smaller. (Here, we consider the lexicographic ordering on $\mathbb{R}^{y-1}$.) Equivalently, if the robustness slack vector is lexicographically minimal, then $p_k > 0$ implies all of the preceding robustness constraints are tight. It thus suffices to show that a lexicographically minimal robustness slack vector exists.

Let $S$ be the set of robustness slack vectors that correspond to feasible solutions. Observe that $S$ is compact. The existence of a lexicographically minimal element in $S$ then follows because any compact subset of $\mathbb{R}^{y-1}$ contains a lexicographically minimal element. To see this last point, note that compactness implies there exists a element with minimum first coordinate. Now, restrict our compact set to this minimum first coordinate and repeat this argument for the second coordinate, and so on. $\qquad\square$

Finally, to prove our main theorem, we need the following technical lemma.

**Lemma A.5.** For all $B > 1$ and $x \in [0, 1]$, the following inequality holds:

$$\frac{1}{B}x - \left(1 + \frac{1}{B-1}\right)^{-1}\left(\left(1 + \frac{1}{B-1}\right)^x - 1\right) \geq 0.$$

*Proof.* The stated inequality is equivalent to

$$\left(1 + \frac{1}{B-1}\right)\frac{1}{B}x + 1 \geq \left(1 + \frac{1}{B-1}\right)^x.$$

Observe that the left-hand side is linear and that the right-hand side is convex. Since the two sides are equal at $x = 0$ and $x = 1$, the desired inequality is true on the interval $[0, 1]$ by Jensen's inequality. $\qquad\square$

Now we are ready to present the proof of the randomized lower bound.

*Proof of Theorem 1.2.* Fix a cost $B$. Given any value of robustness $\gamma > 1$, we give a lower bound for the consistency $\beta$ in the "hard" case $y = 2B - 1$. Assume for now that the LP with parameters $\gamma$ and $\beta$ is feasible. We will derive constraints on $\beta$ in terms of $\gamma$.

By Lemma A.2, we know that there exists a feasible solution $p$ with $p_{y+1} = 0$. By Lemma A.4, there exists a $k \leq y$ such that $p_i > 0$ for $1 \leq i \leq k$ and $p_i = 0$ for all $i > k$. Moreover, the first $k - 1$ constraints of the LP all have 0 slack. In fact, $k \leq B$ always: By our feasibility assumption, $\gamma$ must be at least the optimal competitive ratio $c^* = e/(e-1)$ in the classic setting, since that setting has no consistency constraint. In the classic setting, we can achieve the optimal competitiveness with $k = B$ and have the first $k$ robustness constraints be tight [KMMO94]. Thus, with $\gamma \geq c^*$, the robustness constraints are relaxed, so we must also have $k \leq B$. With these observations and the constraint $p_1 + p_2 + \cdots + p_k = 1$, we can determine the value of $k$ and the probabilities $p_1, \ldots, p_k$.

It is not difficult to see via induction on $k$ (*e.g.*, using the reformulation of Equation (A.3)) that for the first $k - 1$ constraints to each have 0 slack, we must have

$$p_i = \frac{\gamma - 1}{B - 1}\left(1 + \frac{1}{B-1}\right)^{i-1}$$

for each $i$ between 1 and $k - 1$. From $p_1 + p_2 + \cdots + p_k = 1$, it follows that $k$ must be the smallest integer such that

$$\sum_{i=1}^{k} \frac{\gamma - 1}{B-1}\left(1 + \frac{1}{B-1}\right)^{i-1} = (\gamma - 1)\left(\left(1 + \frac{1}{B-1}\right)^k - 1\right) \geq 1.$$

Rearranging the inequality now gives us

$$k = \left\lceil \frac{\log\left(1 + \frac{1}{\gamma-1}\right)}{\log\left(1 + \frac{1}{B-1}\right)} \right\rceil.$$

This choice of $k$, our definition of $p_i$ for $1 \leq i \leq k - 1$, and the constraint $\sum_{i=1}^{k} p_i = 1$ fully determine a feasible solution.

By our assumption that the LP is feasible, this setting of $p_i$ values must also satisfy the consistency constraint. That is,

$$\sum_{i=1}^{k}(B + i - 1)p_i = B + \sum_{i=1}^{k}(i-1)p_i \leq \beta B. \tag{A.4}$$

The remainder of this proof consists of computing the left-hand side of the above for our feasible solution explicitly to obtain a lower bound for the consistency $\beta$.

Applying our explicit formulas for $p_i$ for $1 \leq i \leq k-1$ and the fact $p_k = 1 - \sum_{i=1}^{k-1} p_i$, we compute the sum

$$\sum_{i=1}^{k}(i-1)p_i = \left(\sum_{i=1}^{k-1}(i-1)p_i\right) + (k-1)\left(1 - \sum_{i=1}^{k-1}p_i\right)$$

$$= (\gamma-1)\left(B - (B-k+1)\left(1+\frac{1}{B-1}\right)^{k-1}\right)$$

$$+ (k-1)\left(1 - (\gamma-1)\left(\left(1+\frac{1}{B-1}\right)^{k-1}-1\right)\right)$$

$$= (k-1)\gamma + (\gamma-1)B\left(1 - \left(1+\frac{1}{B-1}\right)^{k-1}\right).$$

It follows from Equation (A.4) that $\beta$ is lower bounded by

$$\beta \geq 1 + \frac{1}{B}\sum_{i=1}^{k}(i-1)p_i$$

$$= 1 + \frac{(k-1)\gamma}{B} + (\gamma-1)\left(1 - \left(1+\frac{1}{B-1}\right)^{k-1}\right).$$

Now, define

$$\Delta k := k - \frac{\log\left(1+\frac{1}{\gamma-1}\right)}{\log\left(1+\frac{1}{B-1}\right)}.$$

After some further computation, we obtain

$$\beta \geq \frac{k\gamma}{B} - \gamma\left(1 - \frac{1}{B}\right)\left(\left(1+\frac{1}{B-1}\right)^{\Delta k}-1\right)$$

$$= \frac{\gamma}{B}\frac{\log\left(1+\frac{1}{\gamma-1}\right)}{\log\left(1+\frac{1}{B-1}\right)} + \gamma\left(\frac{\Delta k}{B} - \left(1+\frac{1}{B-1}\right)^{-1}\left(\left(1+\frac{1}{B-1}\right)^{\Delta k}-1\right)\right).$$

Lemma A.5 lets us bound the terms involving $\Delta k$ from below, and it follows that

$$\beta \geq \frac{\gamma}{B}\frac{\log\left(1+\frac{1}{\gamma-1}\right)}{\log\left(1+\frac{1}{B-1}\right)}.$$

To finish our proof of the lower bound, notice that $B\log(1+1/(B-1)) \to 1$ from below as $B \to \infty$. Hence, the lower bound on $\beta$ approaches $\gamma\log(1+1/(\gamma-1))$ as $B \to \infty$. $\qquad\square$

## B Proof of Theorem 3.2

Now we present our algorithmic result. Although our analysis deals with the case of 2 jobs, it is convenient to describe the algorithm in the general case of $n$ jobs. The algorithm starts by running round robin for a while, then switches to a greedy strategy of processing jobs in the increasing order of the predicted times. If at any point we know $x_i \neq y_i$ for any job $i$, we switch to round robin forever. We use $\mathsf{OPT}_y = \sum_i iy_i$ to denote the $\mathsf{OPT}$ under perfect predictions.

---
`Two-Stage-Schedule`$(y_1, y_2, \cdots, y_n)$:
    At any point, if a job finishes with processing time less or more than its prediction,
        round robin forever.
   *Stage* 1: Round robin for at most $\lambda n \cdot \mathsf{OPT}_y/\binom{n}{2}$ units of time.
   *Stage* 2: Process jobs in predicted order
        (staring from the unfinished job with the least predicted time).
---

The intuition behind the algorithm is simple. On one hand, to ensure robustness, the algorithm switches to round robin when any misprediction is noticed. On the other hand, we ask the algorithm to be $(1 + \lambda)$-consistent. Suppose $y_1 < y_2 < \cdots < y_n$. If the predictions are perfect, then we expect that a consistent algorithm would produce a schedule that finishes the jobs in the correct order, *i.e.*, job 1 finishes first, job 2 second, and so on. In this case, the consistency requirement reduces to

$$\sum_{i>j} d(i,j) \leq \lambda \, \mathsf{OPT}_y, \tag{B.1}$$

where and $d(i,j)$ denotes the amount job $i$ delays job $j$ in this scenario. Observe that when no job is completed, round robin increases each term in the summation at the same rate of $1/n$. Thus, stage 1 of the algorithm would make the inequality (B.1) tight. Then as we can no longer disobey the predictions in the ideal scenario, we switch to the greedy strategy in the second stage. Next, we analyze the performance of the algorithm in the case of two jobs.

We now prove Theorem 3.2

*Proof of Theorem 3.2.* Let $t = 2y_1 + y_2$. To show consistency, assume $x_1 = y_1, x_2 = y_2$, so $\mathsf{OPT} = t$. In stage 1, the algorithm runs round robin for $2\lambda t$ units of time. Observe that job 2 cannot finish before job 1 in this stage: since $\lambda < 1/3$, job 2 can receive at most $(2y_1 + y_2)/3 < y_2$ units of processing time. Consider two cases.

(i) Suppose job 1 finishes in stage 1. Then since two jobs share the same rate,

$$y_1 \leq \lambda t. \tag{B.2}$$

Moreover, in this case. the algorithm runs round robin for $2y_1$ time and finishes job 2 in $y_2 - y_1$ time. Thus, $\mathsf{ALG} = 3y_1 + y_2$, and $\mathsf{OPT} = t$. By (B.2), we have $\mathsf{ALG} \leq (1 + \lambda) \mathsf{OPT}$.

(ii) Suppose job 1 does not finish in stage 1. Then both jobs have been processed for $\lambda t$ units of time at the beginning of stage 2. In stage 2, the algorithm prioritizes job 1. Thus,

$$\mathsf{ALG} = 4\lambda t + 2(y_1 - \lambda t) + (y_2 - \lambda t) = (1 + \lambda) \mathsf{OPT} \tag{B.3}$$

To show robustness, we consider mispredictions, and suppose without loss of generality $y_1 = 1$. Throughout, we let $\epsilon$ to denote an infinitesimal quantity. Notice that if any misprediction is found or job 1 is finished in stage 1, the algorithm is equivalent of round robin and, therefore, achieves $4/3$ competitive ratio that is better than $1 + 1/(1 + 6\lambda)$ for any $\lambda \in (0, 1/3)$, so we are done. We do a case-by-case analysis, assuming in stage 1 no misprediction is detected and both jobs are finished in stage 2. Notice that under the assumptions, $x_1, x_2 \geq \lambda t$, so $\mathsf{OPT} \geq 3\lambda t$.

(i) Suppose job 1 finishes no later than its prediction ($x_1 \leq 1$). We have $\mathsf{ALG} = \lambda t + 2x_1 + x_2$.

   (a) If $x_1 < x_2$, then $\mathsf{OPT} = 2x_1 + x_2$. Since $\lambda t \leq \mathsf{OPT}/3$, we have $\mathsf{ALG}/\mathsf{OPT} \leq 4/3$.

   (b) If $x_1 \geq x_2$, then $\mathsf{OPT} = 2x_2 + x_1$. Observe that setting $x_1 = y_1 = y_2 = 1, x_2 = \lambda t + \epsilon$ maximizes the competitive ratio, and this yields a ratio of $1 + 1/(1 + 6\lambda)$.

(ii) Suppose job 1 finishes later than its prediction ($x_1 > 1$). In this case, the stage 2 starts off by processing job 1 for $y_1 - \lambda t$ unit of time then switching to round robin.

   (a) If job 1 finishes no later than job 2, then we calculate that $\mathsf{ALG} = \lambda t + 3x_1 + x_2 - 1$. If $x_1 < x_2$, then $\mathsf{OPT} = 2x_1 + x_2$, the competitive ratio is at most $4/3$, where the worst case is achieved at $x_1 = 1 + \epsilon$ and we use $\lambda t \leq \mathsf{OPT}/3$. If $x_1 \geq x_2$, then $\mathsf{OPT} = 2x_2 + x_1$. The competitive ratio is bounded by $1 + 1/(1 + 6\lambda)$, where the worst case is achieved when $x_1 = 1 + \epsilon, x_2 = \lambda t + 2\epsilon, y_2 = 1$.

   (b) If job 1 finishes later than job 2, then $\mathsf{ALG} = 1 + x_1 + 3x_2 - \lambda t$. Observe that in this case, it is impossible that $x_2 > x_1$, since job 1 receives more processing than job 2 throughout the schedule. Assume $x_2 \leq x_1$; then the competitive ratio is bounded by $1 + 1/(1 + 6\lambda)$ with the worst case being $x_2 = \lambda t + \epsilon, x_1 = 1$. $\square$