[Reviews · NeurIPS 2020]

Review 1

Summary and Contributions: The paper considers the ski-rental and non-clairvoyant job scheduling problems an online model which mixes predictions with the adversarial setting. The goal is to optimize the trade-off between performance under good predictions (consistency) and performance under bad predictions (robustness). Both problems were given upper bounds in foundational prior work on the model. The present work provides new lower bounds which are tight for the ski-rental problem. They also present a tight algorithm for the special case of non-clairvoyant job scheduling with only two jobs. UPDATE: I have read the rebuttal and other reviews. My score remains the same.

Strengths: The foundational papers on this model [34,42] have gotten a lot of attention and this complements that important work well. The paper is well-written with good presentation.

Weaknesses: No major weaknesses.

Correctness: The claims appear to be correct.

Clarity: Yes.

Relation to Prior Work: Along with citing [18, 15, 35, 37, 39], the authors may want to address the following more closely related paper which mixes the model of [15] with an adversarial setting. Online allocation with traffic spikes: Mixing adversarial and stochastic models Hossein Esfandiari, Nitish Korula, Vahab Mirrokni

Reproducibility: Yes

Additional Feedback: The robustness of the random algorithm in [42] is stated as a function of the budget, but the robustness stated in Theorem 1.2 here is not. Is there significance to the problem with only two jobs aside from being a natural special case to explore theoretically? I am not aware of anything, but it seems like it could be a subroutine for something. If so, explaining this would strengthen the result. Regarding footnote 1: Expected competitive ratio also commonly takes expectation over the randomness in the online model even for deterministic algorithms.


Review 2

Summary and Contributions: The paper considers the problem of designing online algorithms that utilize machine learning predictions for the ski rental and non-clairvoyant scheduling problem. In the learning augmented setup, the consistency of an algorithm is defined to be its competitive ratio given perfect predictions whereas the robustness of an algorithm is defined to be its competitive ratio when the predictions are arbitrarily erroneous. Unlike most recent work in this area, the paper focuses on providing lower bounds on the tradeoffs between robustness and consistency in this framework. In particular, they show that for the ski rental problem the learning augmented algorithms of [42] actually yield the optimum tradeoff between robustness and consistency (for both deterministic and randomized algorithms). The proofs of these lower bounds are non-trivial but not surprising. The paper also shows a lower bound for the non-clairvoyant scheduling problem to minimize weighted completion time although it does not match the algorithms of [42]. For the special case of only 2 jobs, the paper develops a new algorithm that yields a much better tradeoff than [42] and also gives a matching lower bound.

Strengths: This is among the first papers to show lower bounds on the trade-offs between robustness and consistency in learning-augmented algorithms. The techniques employed are non-trivial and lead to some interesting insights - for e.g. that the learning augmented algorithm for non-clairvoyant scheduling is probably not optimal.

Weaknesses: The algorithmic contributions of the paper are a bit weak. In my opinion, the paper will be significantly strengthened if the analysis of the final algorithm could be improved to the case with more than 2 jobs.

Correctness: Yes

Clarity: Yes

Relation to Prior Work: Yes

Reproducibility: Yes

Additional Feedback: Other comments: - In Theorems 1.1 and 1.2, mention “deterministic algorithm” and “randomized algorithm” respectively. Currently although the theorem is titled appropriately, the statement does not specify deterministic / randomized algorithms at all. - In the section of non-clairvoyant scheduling (page 3), the term “makespan” is misused. In typical scheduling literature, makespan is always used to denote the earliest time by which all jobs are scheduled. The quantity measured here is simply called “total completion time”.


Review 3

Summary and Contributions: The authors study performance of classic online algorithms for ski-rental and scheduling with ML advice. The paper proposes algorithms that have guarantees better than the worst-case when the predictions are good and not much worse than it when the advice is bad.

Strengths: Learning augmented algorithms is an important paradigm for designing more practical algorithms for many online problems such as caching, scheduling, routing, etc. The paper proposes simple algorithms with lower bounds for both ski-rental and non-clairvoyant scheduling. The algorithm for scheduling problem is tight.

Weaknesses: The analysis for the ski-rental problem is rather straightforward. Thm 1.1 is immediate. A simple proof is the following: Suppose the prediction is B. Then, to get consistency of 1 + \lambda, you cannot buy after \lambda B. But, if you buy at or before \lambda B, then robustness is 1 + 1/\lambda. Thm 1.2, while offering a more technical proof, does not offer significantly new insights.cIt'll help to add some empirical evaluation of the algorithms. -- Update -- The author's have responded to my questions satisfactorily.

Correctness: I did not carefully read all the theorems and proofs. The claims in the paper look correct.

Clarity: The paper is well written and all the details could be easily followed.

Relation to Prior Work: A comparison with the algorithm proposed in Gollapudi and Panigrahi, ICML 2019, for the single expert case will help differentiate the proposed solutions better.

Reproducibility: Yes

Additional Feedback:

[Author Response · NeurIPS 2020]

We thank all the reviewers for their thoughtful comments and suggestions. We will fix all typos and mechanical errors in the camera-ready and future arXiv versions. Reviewer-specific responses follow.

**Reviewer #1** Thank you for pointing us to the interesting work of Esfandiari, Korula and Mirrokni (2018). We will certainly reference this work in the final version.

Regarding our randomized lower bound for ski-rental, our lower bound as stated in Theorem 1.2 holds in the regime of $B \to \infty$. Note that this exactly matches the upper bound from Purohit et al. A full, precise statement that is in terms of $B$ can be found on line 533 (Appendix A).

As for non-clairvoyant scheduling, we give lower bounds for all numbers of jobs $n$ (Theorem 1.3). Furthermore, any lower bound for $k$ jobs translates into the same lower bound for $n \geq k$ jobs. (Note that one can simply "pad" the input with jobs that takes $0$ time to complete.) Our tight analysis for $n = 2$ shows that the example we find for $2$ jobs is in the hardest one; however, it is not handled optimally by the algorithm of Purhoit et al. Although our focus in this paper is understanding lower bounds, this suggests that there may be more room for future work to improve on the upper bound side, and we indeed suggest an algorithm towards this direction.

**Reviewer #2** Thank you for pointing out the distinction between "makespan" and "total completion time". We will fix that for the full version.

**Reviewer #3** Thank you for pointing out the simple proof of the deterministic lower bound for ski-rental. We will mention this in the full version of this paper.

For randomized ski-rental, we mention that our approach is LP-based (instead of relying on an ad-hoc construction of hard distribution). In general, LP-based arguments (e.g., primal-dual method) are common in obtaining *upper bounds* in competitive analysis. However, they are quite rare in proving (tight) lower bounds. Therefore, we think our proof is technically interesting. We do hope that the strategy can be more broadly applied.

Gollapudi and Panigrahi (2019) considers the setting of having multiple predictions for ski-rental. For the special case of one prediction, their result (Theorem 9) subsumes our deterministic lower bound. However, no (tight) tradeoff is given when randomization is allowed. We will add the discussion in the full version.

[Meta-Review · NeurIPS 2020]

This paper makes progress on a new and important model of algorithm analysis when given predictions. The reviewers appreciated the results and framework.